# Refined Capsid Structure of Human Adenovirus D26 at 3.4 Å Resolution

**DOI:** 10.3390/v14020414

**Published:** 2022-02-17

**Authors:** Vijay S. Reddy, Xiaodi Yu, Michael A. Barry

**Affiliations:** 1Department of Integrative Structural and Computational Biology, The Scripps Research Institute, La Jolla, CA 92037, USA; xyu6@its.jnj.com; 2Department of Internal Medicine, Mayo Clinic, Rochester, MN 55902, USA; barry.michael@mayo.edu; 3Division of Infectious Diseases, Mayo Clinic, Rochester, MN 55905, USA; 4Department of Molecular Medicine, Mayo Clinic, Rochester, MN 55905, USA; 5Department of Immunology, Mayo Clinic, Rochester, MN 55905, USA

**Keywords:** adenovirus structure, protein network, protein–protein interactions, cement proteins, clotting factors

## Abstract

Various adenoviruses are being used as viral vectors for the generation of vaccines against chronic and emerging diseases (e.g., AIDS, COVID-19). Here, we report the improved capsid structure for one of these vectors, human adenovirus D26 (HAdV-D26), at 3.4 Å resolution, by reprocessing the previous cryo-electron microscopy dataset and obtaining a refined model. In addition to overall improvements in the model, the highlights of the structure include (1) locating a segment of the processed peptide of VIII that was previously believed to be released from the mature virions, (2) reorientation of the helical appendage domain (APD) of IIIa situated underneath the vertex region relative to its counterpart observed in the cleavage defective (*ts1*) mutant of HAdV-C5 that resulted in the loss of interactions between the APD and hexon bases, and (3) the revised conformation of the cleaved N-terminal segments of pre-protein VI (pVIn), located in the hexon cavities, is highly conserved, with notable stacking interactions between the conserved His13 and Phe18 residues. Taken together, the improved model of HAdV-D26 capsid provides a better understanding of protein–protein interactions in HAdV capsids and facilitates the efforts to modify and/or design adenoviral vectors with altered properties. Last but not least, we provide some insights into clotting factors (e.g., FX and PF4) binding to AdV vectors.

## 1. Introduction

Human adenovirus D26 (HAdV-D26, or Ad26 for short) causes acute conjunctivitis and a prolonged enteric infection among humans [1]. In addition to differences in the hypervariable regions (HVRs), Ad26 displays a short-shafted fiber comprising eight β-spiral repeats, compared with long-shafted fibers found in archetypal HAdV-C5. Significantly, Ad26 is being used as a vaccine vector against the diseases such as SARS-CoV-2 and AIDS [2,3,4,5,6,7]. A few years ago, we determined the cryo-EM structure of replication-defective HAdV-D26 (RD-HAdV-D26) at 3.7 Å resolution, which showed the conservation in the structural organization of minor/cement proteins among human adenoviruses [7]. With the advent of improvements in data processing and image reconstruction programs, we revisited the dataset that was collected previously and obtained an improved cryo-EM structure at 3.38 Å resolution. Using this higher resolution map, we obtained a refined model for Ad26 capsid by performing real-space refinement using Phenix [8] and model building in Coot [9,10], providing new insights into the structures and organization of minor proteins, IIIa, VI, and VIII, which were not previously considered. The summary of the changes made to the models of individual capsid proteins (CPs) is listed in Table 1. Figure 1 shows the overview of the Ad26 virion structure and organization of various proteins in the AdV capsid.

## 2. Results and Discussion

### 2.1. All Hypervariable Regions (HVRs) of the Hexon Are Ordered in Ad26 Virion

As it has been reported, the overall structure of Ad26 hexon is very similar to that of HAdV-C5 (Ad5) with an RMSD (root-mean-square deviation) of 0.80 Å for 811 pairs of aligned Cα-atoms [7]. Not surprisingly, the significant structural differences between the hexons from Ad26 and Ad5 occur in the HVR loops. It is noteworthy that all HVRs, including HVR1, are ordered in the Ad26 hexon (Figure 2A). Notably, the HVR1 in Ad26 hexon is shorter by 12 a.a., compared with its counterpart in Ad5 hexon [7] (Appendix A). As previously noted, the structural differences that occur in the highly exposed HVRs 5 and 7 appear to correlate well with the observations that Ad26 does not bind coagulation factor X (FX) [11,12,13]. Apart from overall improvements in the quality of hexon models, no specific revisions were made to the hexons in the refined structure of Ad26.

### 2.2. Penton Base and Fiber

The structure of Ad26-PB is also highly conserved, compared with other adenoviruses, with an RMSD of 0.77 Å for 425 pairs of aligned Cα-atoms, with respect to its counterpart in the Ad5 structure (Figure 2B). Even though shorter by 40 a.a., compared with Ad5-PB, the Arg-Gly-Asp (RGD)-containing loop that is known to bind to cell surface integrin molecules is disordered in Ad26-PB. Other differences between the PBs of Ad26 and Ad5 structures occur in the loop (a.a. 139–153) that connects β3 and β4 strands, where there is a six-residue insertion found in Ad26 (Appendix A). In addition, a 12-residue deletion is noted at the N-terminus of Ad26 PB, relative to species-C viruses [7].

We found a 17-residue peptide comprising a.a. 4–20 at the N-terminus (NT) of fiber protein bound tightly at the interface of a pair of PB subunits in the pentameric penton (Figure 2C). The quality of the density for this fiber N-terminal (FNT) peptide is comparable to that of neighboring PBs. Moreover, the FNT-peptide adopts “elbow” shaped conformation that appears to hook the helix at the base of the disordered RGD containing loop, which perhaps is critical for fiber latching onto PB. Notably, even though we observed five copies of FNT-peptide in the Ad26 cryo-EM map, this is due to an artifact of imposing icosahedral fivefold symmetry on the threefold symmetric fiber molecule. As previously inferred, only three out of five copies should correspond to the trimeric fiber molecule [7,14,15,16]. Apart from overall improvements in the quality of PB and FNT models, no specific revisions were made to these models in the refined structure of Ad26.

Ad26 has been reported to utilize sialic acid, CD46, and integrins as receptors. The short-shafted Ad26 fiber is rigid, as reflected by the ability to observe it in cryo-EM studies. This rigidity likely also affects its ability to interact with receptors. In most cases, these interactions are low affinity in the range of µM and are orders of magnitude weaker than Ad5 fiber’s interactions with CAR. While sialic acid and CD46 interactions were expected to be mediated by the Ad26 fiber [17], recent research actually demonstrated that CD46 interactions are surprisingly mediated by the hexon protein [18]. While interactions with sialic acid and CD46 have been somewhat controversial, interactions with integrin appear to be mediated as expected by the Ad26 penton base (PB).

### 2.3. Interactions Involving Protein-IX Are Conserved with Respect to Species-C HAdVs

The Ad26 structure provided the first report of a well-ordered full-length protein-IX (IX) [7] (Figure 3A) that revealed the connection (linker region) between the triskelion and coiled-coil forming regions in HAdVs. Even though the linker region is fully ordered in one (R) of the four structurally distinct IXs-P, Q, R, and S, it clearly demonstrated that the N-terminal triskelion forming regions (a.a. 1–61) are structurally conserved, while the middle linker regions (a.a. 62–93) and C-terminal coiled coils (a.a. 94–134) adopt varied conformations “molded” by the hexameric bases of the hexons [19,20,21] (Figure 3B). In total, 240 copies of the malleable IXs, by displaying 4 distinct conformations, facilitate the formation of the continuous hexagonal lattice comprising triskelions and four-helical bundles (4-HLXB) that represent the nodes (hubs) of the network while the linker regions act as connections between the nodes (Figure 3C). As it was observed previously, the 4-HXLB (coiled coils) is formed by four helices belonging to four structurally unique IXs (P, Q, R, and S) that remarkably come from four different triskelions: three from the same facet and the fourth one from the neighboring facet [19,20]. It is noteworthy that the antiparallel helix, contributed by the IX (P) that comes from the neighboring facet, plays the role of “handshaking” interactions with three other helices in the 4-HLXB and is essential for forming the continuous hexagonal lattice. Moreover, the hexagonal network of IXs interlaces the hexons that form a group of nine (GON) hexons (2–4) but not the peripentonal hexons, throughout the AdV capsid. Notably, a different arrangement of IXs, distinct from the above hexagonal network, has been observed in animal and reptilian adenoviruses [22,23,24], as well as in species-F HAdVs [25,26]. Apart from overall improvements in the quality of IX models, no specific revisions were made to these models in the refined structure of Ad26.

### 2.4. The Appendage Domain of IIIa Is Oriented Differently from Its Precursor Observed in ts1 Mutant of HAdV5

The structure of the majority of the visible Ad26-IIIa comprising N-terminal domain (NTD) (a.a. 4–132) and middle domain (MDLD) (a.a. 133–282) is conserved with respect to its counterpart in mature Ad5-IIIa [27]. However, there is an extra helical domain (a.a. 314–396), termed appendage domain (APD), which is ordered—albeit weakly—in the Ad26 structure but appears to be completely disordered in the mature Ad5-IIIa (Figure 4A). However, the APD has also been reported recently in the structure of a maturation deficient and thermostable Ad5 virion (Ad5-*ts1*) [28]. Although the APD in Ad26 is not well ordered as in Ad5-*ts1*, it is oriented differently relative to its counterpart in the *ts1* virion (Figure 4B). It appears that the APD domain in Ad26-IIIa is rotated by ~120°, compared with its counterpart in the Ad5-ts1 structure, which results in a longer distance (by ~25 Å) between the visible ends of MDLD and APD domains (Figure 4B). It is noteworthy that this reorganization of the APD domain in the Ad26 structure results in the loss of interactions with the contacting hexon subunits. The calculated buried surface area (BSA) between the APD and hexon subunits in the Ad26 structure (838 Å^2^) is reduced by half, compared with a similar calculation of BSA (1554 Å^2^) in the ts1 structure (PDB-ID: 7S78), suggesting the loss of interactions between the APD and hexon bases in the mature Ad26 virions. The BSA values were calculated using ChimeraX [29]. In other words, the association of APD with the hexon subunits is stronger in the *ts1* structure, which agrees with the greater thermostability attributed to the *ts1* virions [30,31]. We surmise that the differences in the organization of APD domains are most likely due to conformational changes associated with the maturation of IIIa, as a.a. sequence of the APD domains are highly conserved between the Ad26 and Ad5 viruses. Based on these observations, we speculate that the mature Ad26 particles are likely to be more stable than mature Ad5 virions, as the APD domain is completely disordered in the latter. However, it is unclear why the APD domain is completely disordered in the structures of mature Ad5 and Ad41 virions [25,26,27]. However, it is noteworthy that IIIa in Ad26 is the smallest in size, composed of 560 a.a. residues, compared with 585 and 579 in Ad5 and Ad41 viruses, respectively. Moreover, the previously assigned residues 283–301, as the continuation of the polypeptide chain of residues 251–282, are now designated as unassigned because of the revised directionality of the helix formed by some of these residues (283–301) and, therefore, are not compatible with this assignment. The remaining a.a. residues 397–560 of Ad26-IIIa, as well as the residues 283–313 connecting the MDLD and APD domains, are disordered.

### 2.5. A Segment of the Processed Peptide (a.a. 140–149) of VIII Is Seen Mediating Interactions between VIII and the N-Terminus of a VI (pVIn) in Hexons 1 and 4

The sequence and structure of VIII are highly conserved among different HAdVs (Appendix A). However, the cleaved peptide comprising a.a. residues 112–157 was not seen before, believed to be released upon proteolytic processing by the AVP [7,32]. In the improved Ad26 structure, we identified 10 residues of the processed peptide (a.a. 140–149) wedged between the a.a.165–172 of VIII and the N-terminus (a.a. 2–8) of the VI (pVIn) (Figure 5). It is noteworthy that in the original Ad26, part of this peptide was assigned to pVIn [7]. It is notable that this short peptide is seen at both the independent locations of VIII on the Ad26 capsid interior. Such an arrangement results in the formation of a three-stranded antiparallel β-sheet involving residues 87–91, 161–171, and 144–146 of VIII, which interacts with the N-terminus (a.a. 2–8) of the closest pVIn (Figure 5A), the remainder of pVIn meanders into the hexon cavity (not shown). Together with the nearby pVIn, the above polypeptides of VIII appear to form a four-stranded β-sheet. In addition, a.a. 140–142 of VIII-U interact with the residues 26–30 in the B subunit of hexon-1, while the same residues in VIII-V interact with the equivalent residues (26–30) in the K subunit of hexon-4. Notably, except for the ordered short segment of the processed peptide (140–149), the rest of the VIII structure is very similar in Ad26 and Ad5 virions (Figure 5B).

### 2.6. The Structures of pVIn Are Highly Conserved

It is now clearly established that VI molecules bind and are released from the hexon cavities [27,28,33,34,35]. In the mature virions of HAdVs, multiple copies of the cleaved N-terminal fragments of VI (pVIn), composed of a.a. 1–33, are seen bound in the hexon cavities [7,27]. In the refined Ad26 structure, the directionality of the polypeptide chains of pVIn in the previous structure was reversed in agreement with the recent Ad5-wt and ts1 virion structures [27,28]. Furthermore, we identified two, two, three, and two copies of pVIn bound in the hexons 1–4, respectively. It is noteworthy that these VI segments were modeled at lower contour levels in varied qualities of densities. The structures of all these polypeptide segments are highly conserved with RMSDs < 1.0 Å for various comparisons between different pairs, most likely influenced by their interactions with the hexon subunits (Figure 6A). Notably, we see conserved stacking interactions between His13 and Phe18 residues in all the segments of pVIn. Strong interactions between pVIn and hexon subunits occur at the entrance of hexon cavity, between residues 6–27 of VI and 26–48 of one hexon subunit (e.g., HX1-B); 16–19; 47–55 of adjacent hexon subunit (e.g., HX1-C) (Figure 6B). As already noted, the residues 2–8 of pVIn interact with a.a. 140–147 of VIII at two locations. This interaction results in the β-sheet complementation by the pVIn (a.a. 2–8) residues that seem to extend the three-stranded β-sheet of VIII into a four-stranded β-sheet (Figure 5). Moreover, it appears that on average 2 copies of VI are bound to each hexon, which amounts to 480 copies of VI packaged in each virion. It is noteworthy that this number is considerably higher, compared with the estimates (359 ± 24) from mass spectrometry-based proteomics analysis of Ad5 virion [36]. It is possible that the above differences in the copy numbers of VI could be due to a different virus (Ad5) was used in the proteomics analysis.

## 3. Conclusions

The overall structure and capsid protein organization of the HAdV-D26 capsid is very similar to that of HAdV-C5. However, the main phenotypic differences between Ad26 and Ad5 virions are likely to occur due to differences in the hexon HVRs as well as in the differences in the receptor binding fiber molecules. The new structural insights from the refined Ad26 structure include (1) identification of a segment of processed VIII peptide (a.a. 140–149), which was previously believed to be released from the mature virions, was found to be interacting with the N-terminus (a.a. 2–8) of VI at both the independent locations (Figure 5); (2) the rearrangement of the APD domain of IIIa in Ad26 structure relative to the corresponding domain in the Ad5-*ts1* structure (Figure 4) [28]; (3) revision of the directionality of polypeptide segments of pVIn (a.a. 1–33) sequestered inside the hexon cavities and their structural conservation that includes the stacking interactions between the His13 and Phe18 residues (Figure 6). Taken together, the new structural information on HAdV-D26 capsid provides opportunities to better understand the aspects of biology that are specific to Ad26 vectors (see below) and overall advance the efforts in designing the improved adenovirus vectors.

One clinically significant difference between Ad26 and species C adenoviruses such as Ad5 is their ability to bind vitamin K-dependent blood clotting factors [11,37]. The hexons of Ad5 and other species C adenoviruses bind coagulation factor X (FX), which appears to partially shield these viruses and protect them from destruction by human liver Kupffer cell after intravenous injection [11,37]. In contrast, Ad26 does not bind FX [11], and it is, therefore, more severely targeted by natural antibodies and complement and destruction by Kupffer cells [38]. The effect of this interaction is profound, as evinced by our recent observation that Ad5 mediates 1000-fold higher transduction of the liver than Ad26 after intravenous injections in CD46 transgenic mice [39].

This differential FX binding may also have a considerable effect on vaccine and vector safety and side effects [40,41,42,43]. Thrombosis and thrombocytopenia have been reported after the COVID-19 vaccination the Ad26.COV2.S COVID-19 vaccine (Johnson & Johnson/Janssen) [40,44,45,46,47,48,49,50,51,52,53,54,55]. While this COVID-19 vaccine has been injected intramuscularly, some fraction of injected vaccine will leak into the blood, thereby potentially inducing effects on the clotting system. Similar to most interactions of Ad26 with receptors and proteins, the binding of blood clotting factor, platelet factor 4 (PF4), also has a low affinity with binding constants for chimpanzee adenovirus vector (ChAdOx1), Ad5, and Ad26 of 0.661, 0.789, and 0.301 µM, respectively [56]. This binding has a 1000-fold lower affinity than the 0.000229 µM binding of FX to Ad5 [37]. The computational modeling studies reported in [56] identified PF4 binding amino acid residues in hexon, including E440 and D442. These residues are notably in close proximity to or overlap FX binding residues in Ad5 (e.g., E424 and T425) [13] (Figure 7).

How FX and PF4 might interact with different adenoviruses may also be strongly driven by their native concentrations in the blood or tissue. The median concentration of PF4 in human blood is 7 ng/mL [57]. The concentration of FX is approximately 8 µg/mL [11]. Therefore, the 1000-fold higher concentration and 1000-fold higher affinity of FX over PF4 may drive FX to coat species C Ads to limit the binding of PF4. In contrast, adenoviruses that do not bind FX, e.g., Ad26, would not be shielded by FX and could be more easily bound by PF4 and may increase the likelihood of the production of anti-PF4 antibodies. Whether there is competition between FX and PF4 and whether FX binding mitigates the risks of thrombotic thrombocytopenia remains to be determined. There are of course many more proteins and cells that could bind different adenoviruses beyond FX and PF4. These interactions are as diverse as the genomes of adenoviruses and likely contribute to AdV biology, efficacy, and safety as therapeutics and vaccines.

If PF4 binding by Ad26 is indeed involved in the thrombotic thrombocytopenia side effects observed during COVID-19 vaccinations, there is perhaps a simple solution to the problem. Do not inject the vaccine into the muscle. Instead, apply the vaccine by the intranasal or another mucosal route where Ad26 is much less likely to encounter the blood and PF4. Of course, other routes have their own unique side effects. However, given that the route of entry by SARS-CoV-2 is intranasal and respiratory, injecting a vaccine to counter this pathogen into the muscle may not be the best strategy.

## 4. Methods

The details of the production of RD-HAdV-D26 virions, cryo-EM sample preparation, and data collection were previously described in detail [7].

### 4.1. Improved Cryo-EM Data Processing Structure Determination

A total of 2017 micrographs were obtained after performing whole-frame alignment and correcting for beam-induced motion, using MotionCorr program [58] were loaded into the cisTEM program [59]. After performing the CTF correction using the CTFfind program [60] within cisTEM, we automatically picked 37,026 virus particles using the maximum and characteristic radius of 400 Å and a threshold peak height of 2. These particles were subjected to reference-free 2D classification, followed by 3D refinement as a single class that included all the picked particles, using the map from the polyalanine model of previously deposited Ad26 structure (PDB-ID:5TX1) as the reference volume. This step resulted in a map of 3.5 Å resolution. Subsequently, we performed 3D classification of the dataset as two classes and selected the best (one) of the classes that included 99.1% (36,706) particles. This was followed by another round of 3D classification as 2 classes. The class chosen for the final round of 3D refinement contained 30,834 particles that resulted in a 3.38 Å resolution map, which after sharpening with a B factor of −90 Å^2^, was used for model building in Coot. The details of cryo-EM data statistics are shown in Appendix A. The FSC plot of Ad26 cryo-EM reconstruction is shown in Appendix A.

### 4.2. Model Building and Refinement

The existing model of the Ad26 structure (PDB-ID: 5TX1) was positioned into the icosahedral asymmetric unit of the newly reconstructed 3D map using chimera [61]. Then, manually adjusted the model wherever the fit to the density was not perfect using the “density fit analysis” tool in Coot [10]. Significantly, however, we corrected the directionality (polarity) of the N-terminal fragment of pre-protein VI (pVIn), in agreement with the recent reports [27,28]. Furthermore, we adjusted the model of the APD domain (314–396) of IIIa by the structural superposition of the well-ordered APD domain in the ts1 structure [28]. Additionally, the previously assigned residues 283–301 are now designated as unassigned because the revised directionality of the helix, formed by some of these residues (283–301), is not consistent with the continuation of the polypeptide of residues 251–282. Moreover, significantly, we built a short segment of residues (140–149) of the protease cleaved fragment (112–157) of VIII into the density seen at both the structurally independent locations. Lastly, we performed real-space refinement of the above model into the density using Phenix [8]. Various structural analyses and generation of figures were carried out using Chimera and ChimeraX [29,61]. The details of model composition and quality statistics are also shown in Appendix A. A bar plot of the map-to-model correlation coefficient (CC) values for individual chains in the Ad26 model is shown in Appendix A.

## Figures and Tables

**Figure 1 viruses-14-00414-f001:**
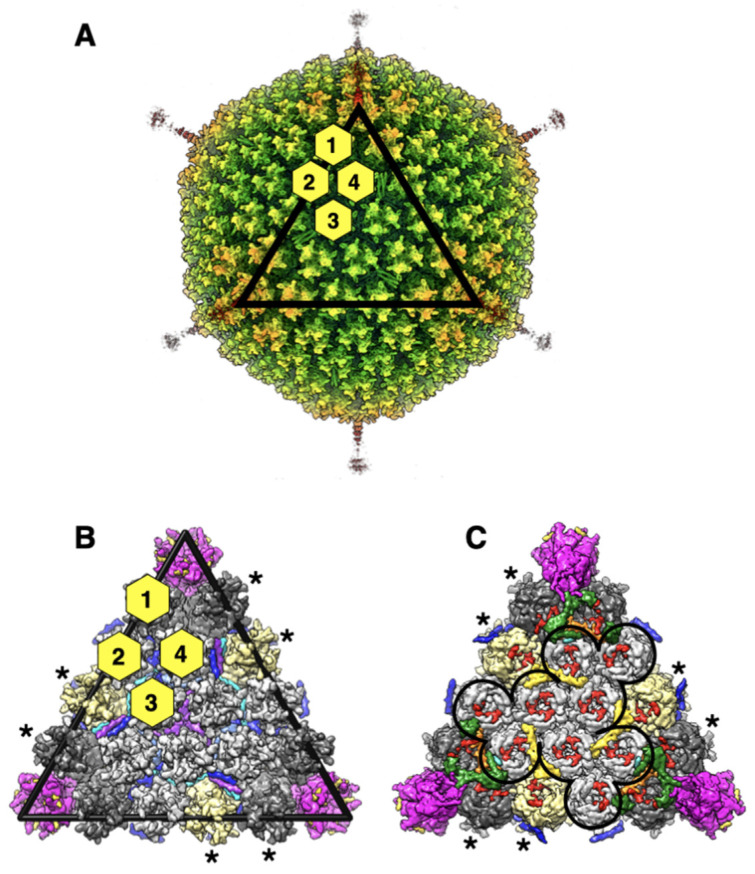
Overall structure and organization of HAdV-D26 (Ad26) capsid: (**A**) illustration of the radially color-coded surface of the cryo-EM reconstruction of Ad26 virion, shown as a view down the icosahedral threefold axis. A rainbow color gradient from blue to red was used to represent the regions of the map between the radii 300 Å and 500 Å, respectively. The icosahedral facet is identified by the black triangle and the four structurally distinct hexon positions are distinguished by the yellow hexagons labeled 1–4; (**B**) a zoomed-in view of the icosahedral facet seen from the outside of the virion. Hexons are depicted in surface representation and those that belong to a group of nine hexons (GONs) are shown in light gray, while the peripentonal hexons (PPHs) are colored in dark gray. The hexons that belong to neighboring facets are identified by asterisks and/or shown in khaki color. The pentamers of penton base (PB), located at the icosahedral vertices are shown magenta. The short fiber N-terminal (FNT) peptides (a.a. 4–20), bound at the interfaces of PB subunits, are shown as gold-colored surfaces. The triskelion and 4-HLXB structures formed by the 12 protein-IXs-3 copies of 4 structurally distinct IXs, identified as P, Q, R, and S, present in an icosahedral facet, are shown in different colors blue, light blue, cyan, and purple, respectively; (**C**) a vertically flipped view of panel B, showing the inside of the facet. The outline of the GON structure is represented by the black line. The surface representation of ordered regions of IIIa, VIII-U, and VIII-V and VI are shown in dark green, orange, yellow, and red colors, respectively. The model fragments corresponding to islands of unassigned densities are shown as turquoise surfaces. The C-terminal helix of the IX(P) can be seen in blue.

**Figure 2 viruses-14-00414-f002:**
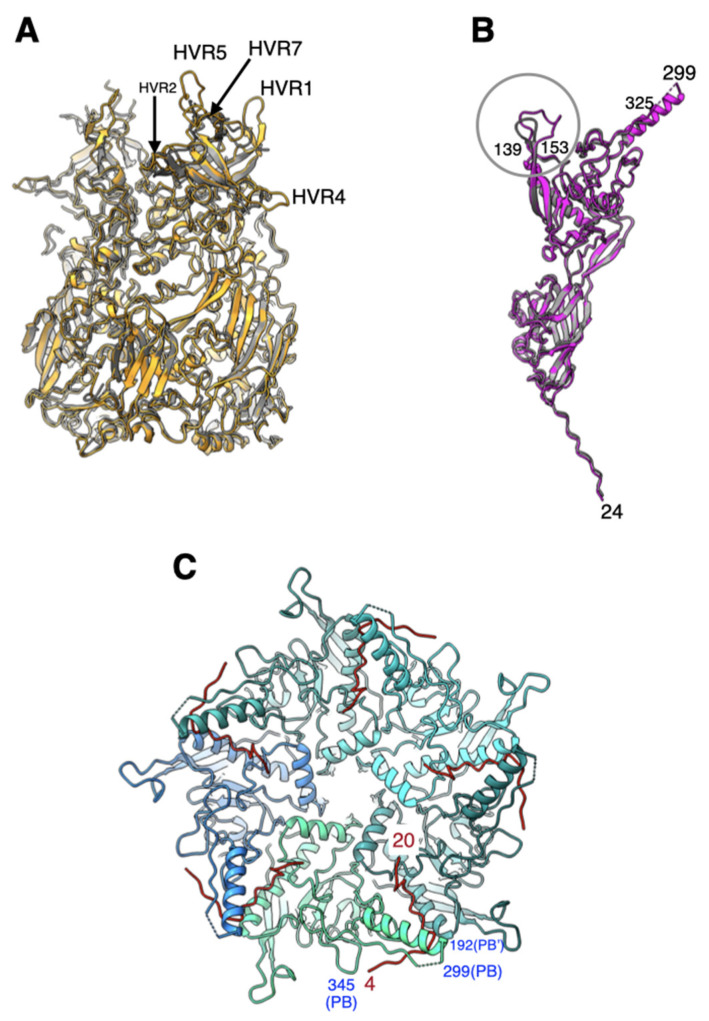
Structural similarities and differences between the major capsid proteins of Ad26 and Ad5 viruses: (**A**) superposition of peripentonal hexons from Ad26 (orange) and Ad5 (gray). Few selected HVR loops ordered in the Ad26 structure are identified; (**B**) superposition of penton base (PB) subunits from Ad26 (magenta) and Ad5 (gray) structures. Major structural differences occur in the loop containing residues 139–153, identified by a circle, where there exists a six-residue insertion in Ad26-PB. The RGD containing loop between the residues 299–325 is disordered; (**C**) An illustration showing the “elbow” shaped fiber N-terminal (FNT) tails (in red) interacting with the PB subunits, which are shown as ribbons in different shades of blue. Few selected PB residues are labeled in blue, while the residues of the FNT peptides are indicated in red.

**Figure 3 viruses-14-00414-f003:**
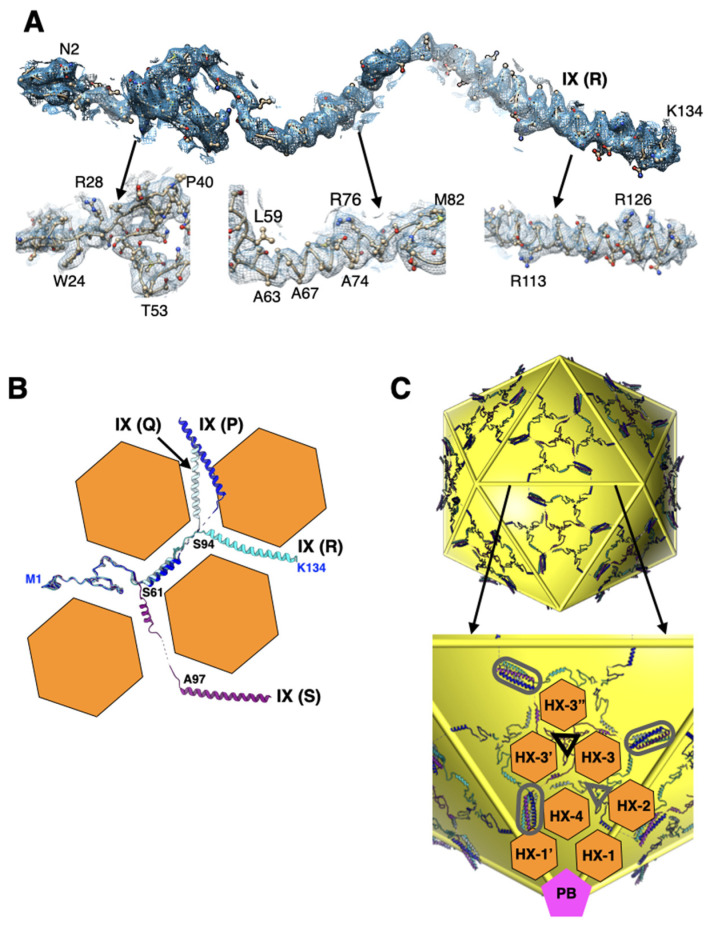
Structure and organization of IX: (**A**) fit of the density for the fully ordered protein-IX (R). Shown below are the zoomed-in views of different regions indicated by the arrows. Few representative residues are labeled; (**B**) superposition of four structurally distinct IXs—P, Q, R, and S—which are shown in colors blue, light blue, cyan, and purple, respectively. The triskelion forming region consists of a.a. 1–61 and the coiled-coil region is formed by residues 94–134, while the in-between residues (62–93) form the linker region. The orange hexagons represent the shape of the bases of major capsid protein, hexon, that appear to mold the conformation of IXs by bending them ~120° at the locations, identified by residues labeled in black, where they contact the vertices of the hexagons. However, the antiparallel helix contributing IX (P) (in blue) appears to deviate from the above angular constraints imposed by the hexon bases; (**C**) the extended protein network shows the overall organization of 240 IXs on the surface of an icosahedron shown in yellow (top). Shown at the bottom is a zoomed-in view of the IX network within an icosahedral facet. The structurally distinct IXs are color coded as in panel B. The location of triskelions and coiled coils (4-HLXB) are identified by triangles and elongated ellipses, respectively. The gray and black colored triangles identify the triskelions located at the quasi (local) threefold and strict threefold axes of symmetry, respectively. The locations of selected hexons and PB are identified by orange hexagons and magenta-colored pentagons, respectively.

**Figure 4 viruses-14-00414-f004:**
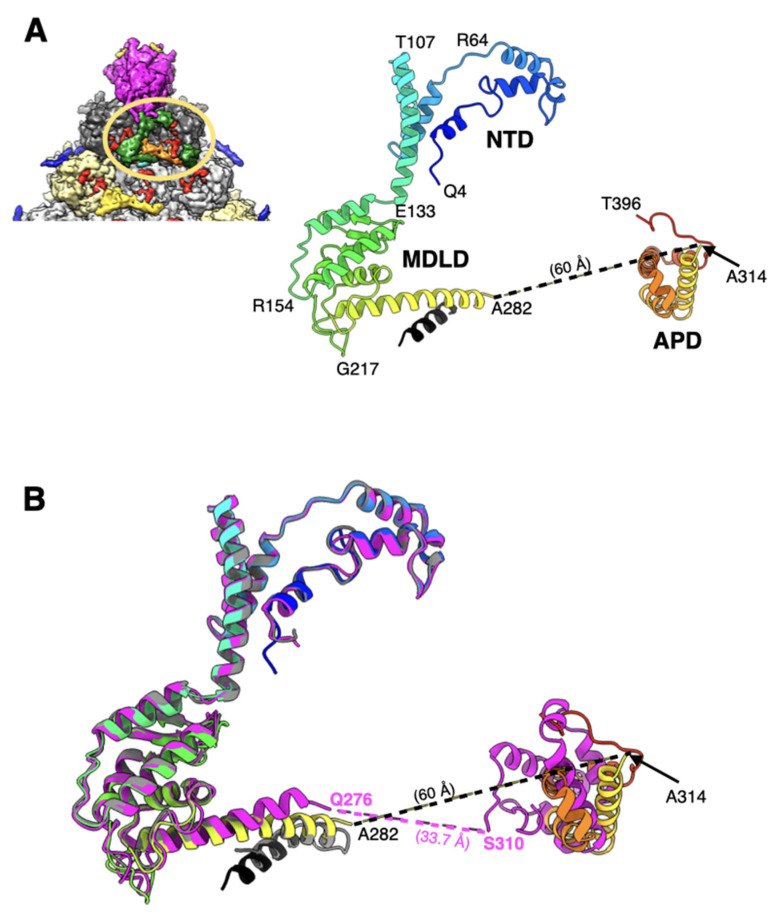
Structure and organization of IIIa: (**A**) shown on the top left is a portion of the icosahedral facet seen from the inside of the Ad26 virion (see Figure 1C for details), indicating the interior location of IIIa represented by green colored surface, and encircled by the orange-colored oval. Illustrated on right is a ribbon diagram of IIIa depicted in rainbow color gradient showing the structure of ordered regions. Different domains of IIIa, along with a few representative residues along the polypeptide chain, are identified. The helix shown in black was built in an unassigned island of density adjacent to the helix formed by a.a. 251–282; (**B**) superposition of ordered regions of IIIa from Ad26 (rainbow gradient), from Ad5-ts1(magenta; PDB-ID: 7S78) and from mature Ad5 (gray; PDB-ID: 6B1T). Even though the structures of NTD and MDLD domains are virtually superimposable in the Ad26 and Ad5-ts1 structure, the orientation of the APD domains seems to be different. The distances between the visible ends of MDLD and APD domains in Ad26 and Ad5-ts1 structures are indicated. Notably, the APD domain is completely disordered in the mature Ad5 structure and the unidentified helix (shown in black), which is closely associated with Ad26-IIIa overlapping with the helix formed a.a. 291–301 of mature Ad5-IIIa. However, the directionality of these helices appears to be the opposite.

**Figure 5 viruses-14-00414-f005:**
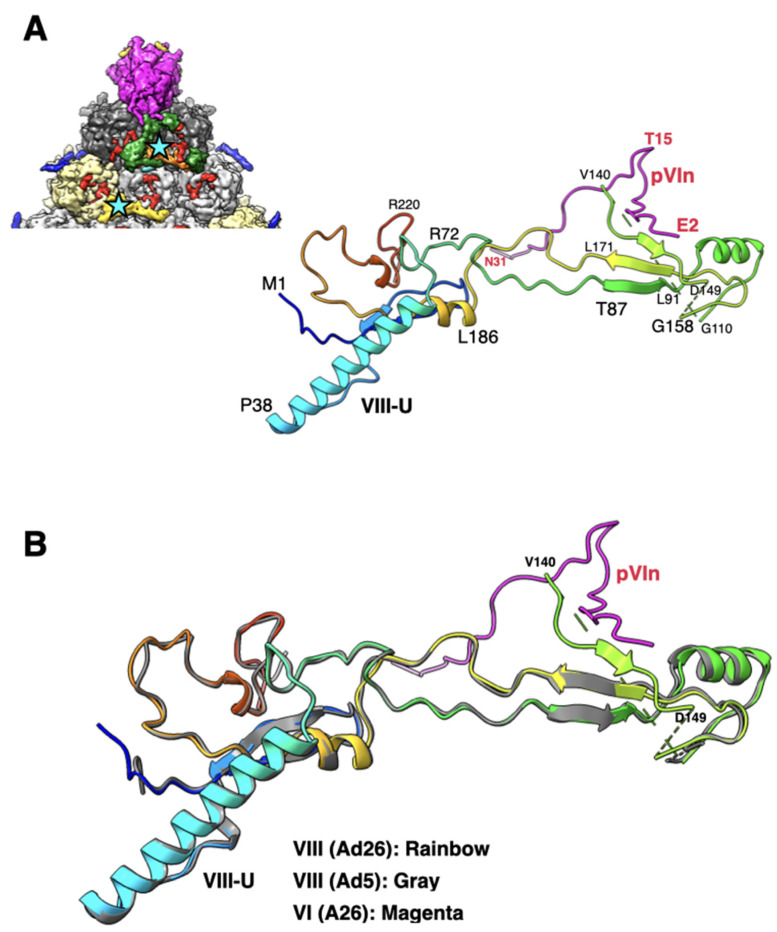
Structure and association of VIII with pVIn: (**A**) shown on the top left is a portion of the icosahedral facet seen from the inside of the virion (see Figure 1C for details), identifying the interior locations of two structurally independent VIII molecules, which are shown in orange (VIII-U) and yellow (VIII-V) colored surfaces and identified by cyan-colored stars. Shown on the right is a ribbon diagram of the VIII-U molecule, depicted in rainbow colors, interacting with pVIn, whose polypeptide backbone is shown in purple. Few selected residues of VIII and VI are labeled in black and red colors, respectively; (**B**) superposition of VIII (U) molecules from Ad26 (rainbow colors) and Ad5 (gray; PDB-ID: 6B1T) structures showing the overall structural similarity with an RMSD of 0.61 Å. However, the residues 140–149 are disordered in the Ad5 structure. The structure of pVIn is shown in purple.

**Figure 6 viruses-14-00414-f006:**
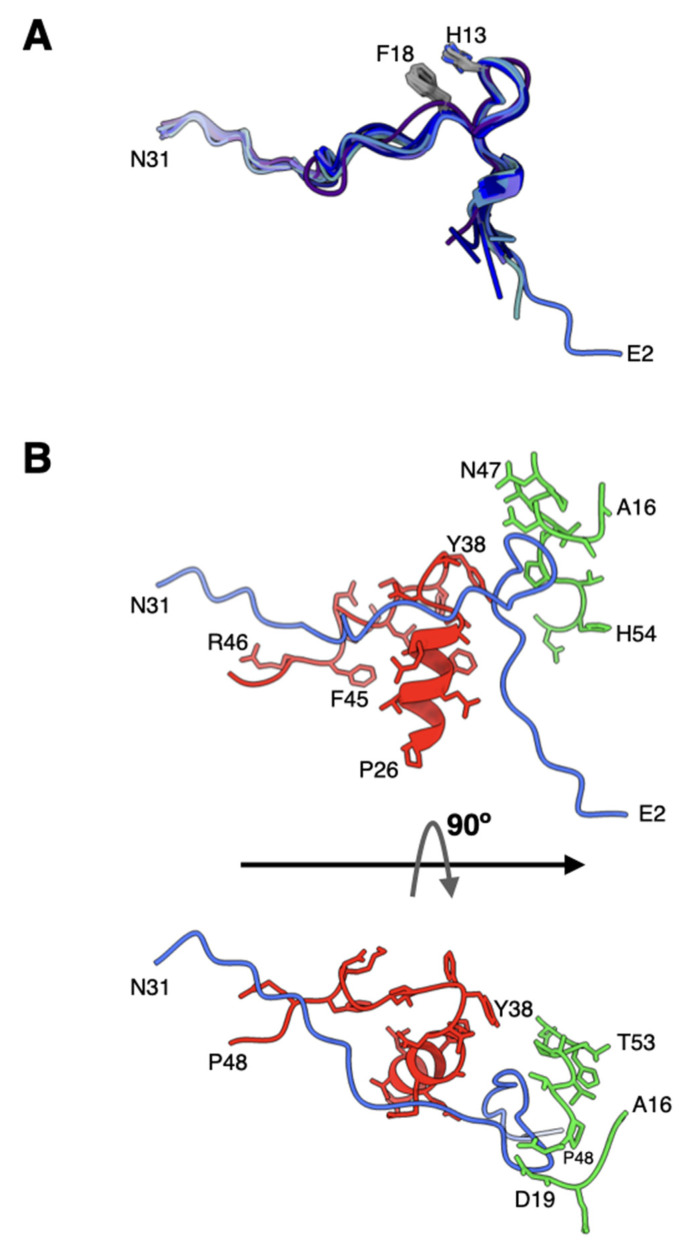
Structure and association of pVIn with hexon: (**A**) superposition of different N-terminal fragments (a.a. 2–31) of VI (pVIn) that are observed in the hexon cavities of Ad26 structure. The structure of pVIn in Ad26 virion appears to be well conserved, likely influenced by its interaction with hexon subunits. The His13 and Phe18 residues that are involved in stacking interactions are identified; (**B**) interaction of pVIn with the selected regions from two hexon subunits, depicted in red and green colors, at the entrance of hexon cavity. Shown at the bottom is a 90° rotated view of the top.

**Figure 7 viruses-14-00414-f007:**
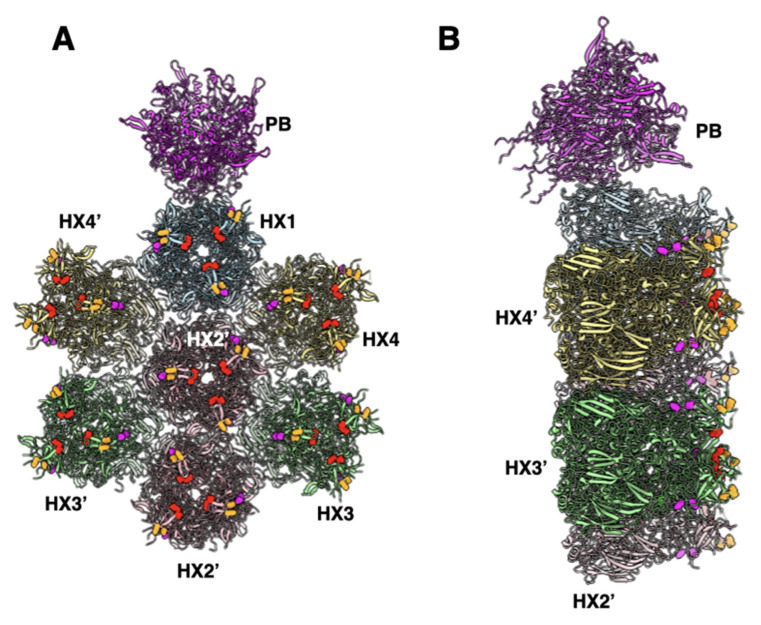
The location of FX and PF4 binding residues in the Ad5 structure (PDB-ID: 6B1T): (**A**) a top view showing a group of contiguous hexons identifying the FX and PF4 binding residues. The residues E424 and T425 that are suggested to bind to the GLA domain of FX [13] are shown in the red spheres, while a few of the predicted PF4 binding residues [56]-D290, D292 shown in magenta and E440 and D442 are shown in orange; (**B**) a side view of panel A showing the relative radial disposition of putative FX and PF4 binding residues on hexons.

**Table 1 viruses-14-00414-t001:** Specific changes made to the CPs in the refined model of Ad26 virion.

Protein	Residue Changes	Remarks
Hexon	None	N/A
PB	None	N/A
Fiber	None	N/A
IX	None	N/A
IIIa	283–301; 314–396 (APD)	Directionality of a.a. 283–301 is reversed and is now designated as unassigned. The APD domain is adjusted and remodeled.
VIII	140–149	A segment of processed VIII peptide (a.a. 140–149) is identified for the first time.
VI	2–31 (pVIn)	Directionality of pVIn chains is reversed and remodeled.

## Data Availability

The cryo-EM map and coordinates of the refined Ad26 virion have been deposited in EMDB and PDB with the accession codes EMD-25786 and 7TAU, respectively.

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
