# Peer review of "Refined Capsid Structure of Human Adenovirus D26 at 3.4 Å Resolution"

_viruses, 2022, doi:10.3390/v14020414_

Round 1

Reviewer 1 Report

Overall, the refined model shows to gain resolution compared to the 2017 model. Using new data analysis and modeling technology novel insights on the Ad26 capsid structure have advanced.

Comments and suggestions:

  • Introduction, line 36-37, the authors refer to Ad26 without the notification these are replication incompetent adenoviruses. Please add this information to the text.
  • Figure 1A, the authors state the radially color-coded view is down the 3-fold axis, however it seems that the center of the radially color-coded view is in the 5-fold axis. It might be the wording but please check.
  • Results and Discussion, 2.3. Interaction involving protein IX are conserved with respect to species-C HAdVs. The authors use various ways to describe protein IX (molecule of IX, IX-molecules, IX molecule, IX (P) molecule, IXs. Please reduce the variety how to refer to protein IX. (also check and align to figure 3). In the subsequent paragraphs the authors use IIIA.
  • Results and Discussion, 2.4. The appendage domain of IIIa is oriented differently from its precursor oserved in ts1 mutant of HAdV5.
    The authors describe the differences of protein IIIa with respect to the APD domain compared to Ad5 and AD5-ts1. Here I favor the authors to speculate more on the impact of the thermostability/capsid stability of Ad26 particles. The authors limited their reasoning to Ad5-ts1.
  • Results and Discussion, 2.6. The structures of pVIn are highly conserved. The authors point out in line 227-230 that the amounts of VI packaged in each virion found by re-analysis of the cryo-electron microscopy dataset is higher that earlier observed in spectrometry-based proteomics. Can the authors reason why this difference can be expected? 
  • Results and Discussion, 2.7. In vivo Biology and Clotting Effects by Ad26. This paragraph does not connect with the refined model described in the earlier paragraph. The described information can be used in a condensed form in the conclusion section.

Conclusion:

  • In general, it would be good to highlight what information has been acquired having the new refined model compared to the model published earlier in 2017 (DOI: 10.1126/sciadv.1602670)). Only two sentences describe the conclusion of paragraph 2.1 to 2.6.
  • The paragraph describing FX binding (lines 265 -273) is not in balance with the data discussed in the manuscript. Here I would expect that the authors provide insight on space and location of the potential PF4 binding to Ad26, using the information of the refined model. The conclusion is predominantly about data that is not discussed in the results and discussion section.

Author Response

Reviewer-1

  1. Introduction, line 36-37, the authors refer to Ad26 without the notification these are replication incompetent adenoviruses. Please add this information to the text.

We now indicated that we determined cryo-EM structure of replication-defective HAdV-D26 (Line: 45).

  1. Figure 1A, the authors state the radially color-coded view is down the 3-fold axis, however it seems that the center of the radially color-coded view is in the 5-fold axis. It might be the wording but please check.

We apologize for the confusion. The virion surface was radially color-coded and shown as a view down the 3-fold axis, which we now mentioned (line: 302). For the record, the radial color-coding is independent of how the surface is shown (i.e.,  down 3-fold vs. 5-fold). The 5-fold vertices of an icosahedron are naturally at higher radial distance than the icosahedral 3-fold facets, therefore color-coded as such.

  1. Results and Discussion, 2.3. Interaction involving protein IX are conserved with respect to species-C HAdVs. The authors use various ways to describe protein IX (molecule of IX, IX-molecules, IX molecule, IX (P) molecule, IXs. Please reduce the variety how to refer to protein IX. (also check and align to figure 3). In the subsequent paragraphs the authors use IIIA.

We now tried to keep the homogeneous nomenclature as protein-IX or simply IX. The IX(P), IX(Q), IX(R ) or IX (S) are used to distinguish the four structurally unique molecules.

Not sure we understand last comment “In the subsequent paragraphs the authors use IIIA”

  1. Results and Discussion, 2.4. The appendage domain of IIIa is oriented differently from its precursor observed in ts1 mutant of HAdV5. The authors describe the differences of protein IIIa with respect to the APD domain compared to Ad5 and AD5-ts1. Here I favor the authors to speculate more on the impact of the thermostability/capsid stability of Ad26 particles. The authors limited their reasoning to Ad5-ts1.

We now added a sentence – “Based on these observations, we speculate that the mature Ad26 particles are likely to be more stable than Ad5 virions as the APD domain is completely disordered in the latter” (Lines: 145-147). This is also in line with our suggestion in the first Ad26 structure paper.

  1. Results and Discussion, 2.6. The structures of pVIn are highly conserved. The authors point out in line 227-230 that the amounts of VI packaged in each virion found by re-analysis of the cryo-electron microscopy dataset is higher than earlier observed in spectrometry-based proteomics. Can the authors reason why this difference can be expected?

It is possible that the differences in the VI copy number could be due to a different virus (Ad5) was used in mass-spectrometry based proteomics analysis. We now added a sentence “It is possible that the above differences in the copy numbers of VI could be due to a different virus (Ad5) was used in the proteomics analysis” (lines:194-196).

  1. Results and Discussion, 2.7. In vivo Biology and Clotting Effects by Ad26. This paragraph does not connect with the refined model described in the earlier paragraph. The described information can be used in a condensed form in the conclusion section.

We agree with reviewer’s suggestion. We now removed the section on “In vivo biology and clotting effects by Ad26” and condensed the information in the conclusions section.

  1. Conclusion: In general, it would be good to highlight what information has been acquired having the new refined model compared to the model published earlier in 2017 (DOI: 10.1126/sciadv.1602670)). Only two sentences describe the conclusion of paragraph 2.1 to 2.6.

We agree with reviewer’s suggestion. We now expanded the conclusions based on new structural findings.

  1. The paragraph describing FX binding (lines 265 -273) is not in balance with the data discussed in the manuscript. Here I would expect that the authors provide insight on space and location of the potential PF4 binding to Ad26, using the information of the refined model. The conclusion is predominantly about data that is not discussed in the results and discussion section.

Overall, we agree with the reviewer’s observation. However, given the general interest on the clotting effects associated with the Ad-vector derived COVID-19 vaccines, we argue to keep the text in the conclusion section. As we indicated above, we removed the results section on “In vivo biology and clotting effects by Ad26”. We appreciate the special consideration in this regard.

Reviewer 2 Report

Adenoviruses are a large group of human dsDNA viruses that cause various diseases, and are also used as vectors for oncolytic therapies and vaccines (most notably several COVID-19 vaccines). In this manuscript Reddy et al present an improved analysis of an existing data set that in 2017 allowed them to determine the structure of human adenovirus D26. This is the virus that is used as the vector for both the Janssen and Sputnik COVID-19 vaccines, and new insights into its structure are thus highly welcome public information. The authors are experts at adenovirus structure and the manuscript is well written with clear figures. I am thus overall positive to publication of this manuscript - but ONLY after significant changes to the presentation.

Major point: The minute improvement in the resolution upon reprocessing means that most features in the new map and model will be identical to the old ones. However, the authors are not clear about what is new/different in the new model compared to the old one. I am thus left with the feeling that several structural features presented and discussed may have already been there in the old model (as an example: were all the HVR loops built already in the old model, or is this an addition in the new model?). This must be improved so that it is clear for each section of results if the authors present new features of the improved model, or are discussing features that were already there in the old model but in the light of new literature (which would also be fine if clearly stated). I would strongly suggest that the authors make a table that lists changes/additions to the new model compared to the old one (i.e. what residues were additionally built, or changed like 283-301 of IIIa). It should also be more clearly presented in each section of the results text.

Minor points:

  1. It is said that the 140-149 fragment of VIII mediates an interaction between VIII and pVIn. It is described how this works on the VIII side (beta sheet complementation) but there is no description of the nature of the interaction between 140-149 and pVIn (i.e. which residues interact and how), neither in the text nor in the figure. That should be added.
  2. Line 87: "weaker that" should be "weaker than"?
  3. Line 215: “released” should be “release” or “are released”?
  4. Section 2.4: Is the APD cleaved during maturation? If so, are there any residues /does the different orientation explain why one can see it in Ad26 and not Ad5/Ad41?
  5. Section 2.7: Although in “results and discussion”, this section is a pure literature survey with no presentation of results from the improved Ad26 structure. It should either be rewritten to clearly refer to the reprocessed improved structure, or perhaps rewritten to be included in “Conclusions”? I appreciate the general interest in this topic and am not suggesting to remove it completely. But my understanding of “results and discussion” is that it should only contain discussion that is related to results, not pure speculation based on literature.

Author Response

Reviewer-2

  1. Major point: The minute improvement in the resolution upon reprocessing means that most features in the new map and model will be identical to the old ones. However, the authors are not clear about what is new/different in the new model compared to the old one. I am thus left with the feeling that several structural features presented and discussed may have already been there in the old model (as an example: were all the HVR loops built already in the old model or is this an addition in the newmodel?). This must be improved so that it is clear for each section of results if the authors present new features of the improved model or are discussing features that were already there in the old model but in the light of new literature (which would also be fine if clearly stated). I would strongly suggest that the authors make a table that lists changes/additions to the new model compared to the old one (i.e., what residues were additionally built, or changed like 283-301 of IIIa). It should also be more clearly presented in each section of the results text.

We thank the reviewer for the comments and suggestions. We would like to point out that it is clearly stated in the abstract what features are new in the refined Ad26 model compared to the original structure as well as in the results sections. However, as per the reviewer’s suggestion, we now included a table (Table 1) listing the changes/modifications made in the refined structure and also indicated in the text wherever there are no specific changes made in the models of respective proteins.

Minor points:

  1. It is said that the 140-149 fragment of VIII mediates an interaction between VIII and pVIn. It is described how this works on the VIII side (beta sheet complementation) but there is no description of the nature of the interaction between 140-149 and pVIn (i.e., which residues interact and how), neither in the text nor in the figure. That should be added.

Thanks for the suggestion. We now added a sentence indicating the role played by residues 2-8 of pVIn in the b-sheet complementation - As already noted, the residues 2-8 of pVIn interact with a.a. 140-147 of VIII at two locations. This interaction results in the b--sheet complementation by the pVIn (a.a. 2-8) residues that seem to extend the 3-stranded b-sheet of VIII into a 4-stranded b--sheet (Fig. 5). (Lines: 188-191).

Of note, the b-sheet complementation involves mainly the backbone interactions between the carbonyl and amide groups, does not involve side chains.

  1. Line 87: "weaker that" should be "weaker than"

Thanks for catching it. Changed as suggested.

  1. Line 215: “released” should be “release” or “are released”?

Thanks for catching it. Changed as suggested.

  1. Section 2.4: Is the APD cleaved during maturation? If so, are there any residues/does the different orientation explain why one can see it in Ad26 and not Ad5/Ad41?

No. The APD domain is composed of residues 314-396. The protease cleavage of IIIa occurs at a.a. 542 (Ad26) / 570 (Ad5). It is difficult to explain with certainty the reasons behind the order/disorder phenomenon seen in the structures of similar viruses.  However, it is worth pointing out that IIIa has fewer residues (560) in Ad26, compared to 585 and 579 in Ad5 and Ad41 respectively. Of note, the APD domain is ordered in the ts1 mutant of Ad5 virion. We now added this detail to the results and discussion. (Lines:149-150)

  1. Section 2.7: Although in “results and discussion”, this section is a pure literature survey with no presentation of results from the improved Ad26 structure. It should either be rewritten to clearly refer to the reprocessed improved structure, or perhaps rewritten to be included in “Conclusions”? I appreciate the general interest in this topic and am not suggesting to remove it completely. But my understanding of “results and discussion” is that it should only contain discussion that is related to results, not pure speculation based on literature.

We now removed the section on “In vivo biology and clotting effects by Ad26” and condensed that information in the conclusions section.

Round 2

Reviewer 1 Report

No comments, the authors have adapted the manuscript where needed.